# Effect of Ultrasonic Surface Impact on the Fatigue Properties of Ti3Zr2Sn3Mo25Nb

**DOI:** 10.3390/ma13092107

**Published:** 2020-05-02

**Authors:** Zhangjianing Cheng, Xiaojian Cao, Xiaoli Xu, Qiangru Shen, Tianchong Yu, Jiang Jin

**Affiliations:** 1College of Civil Engineering, Tongji University, Shanghai 200092, China; 1911382@tongji.edu.cn; 2School of Transportation & Civil engineering, Nantong University, Nantong 226019, China; xuxiaoli@ntu.edu.cn (X.X.); shenqr@ntu.edu.cn (Q.S.); yu_tianchong@163.com (T.Y.); jin.j@ntu.edu.cn (J.J.)

**Keywords:** fatigue, surface modification, titanium alloy, nanocrystal

## Abstract

The effect of nano grain surface layer generated by ultrasonic impact on the fatigue behaviors of a titanium alloy Ti3Zr2Sn3Mo25Nb (TLM) was investigated. Three vibration strike-numbers of 24,000 times, 36,000 times and 48,000 times per unit are chosen to treat the surface of TLM specimens. Nanocrystals with an average size of 30 nm are generated. The dislocation motion plays an important role in the transformation of nanograins. Ultrasonic surface impact improves the mechanical properties of TLM, such as hardness, surface residual stress, tensile strength and fatigue strength. More vibration strike numbers will cause a higher enhancement. With a vibration strike number of 48,000 times per square millimeter the rotating-bending fatigue strength of TLM at 10^7^ cycles is improved by 23.7%. All the fatigue cracks initiate from the surface of untreated specimens, while inner cracks appear after the fatigue life of 10^6^ cycles with the ultrasonic surface impact. The crystal slip in the crack initiation zone is the main way of growth for microcracks. Crack cores are usually formed at the junction of crystals. The stress intensity factor of TLM titanium alloy is approximately 7.0 MPa·m^1/2^.

## 1. Introduction

As a kind of advanced multifunctional material, biomedical materials can be used to diagnose, cure, repair or replace human tissues, organs or enhance their functions. Their unique efficacy is irreplaceable by drugs. In view of the high strength-to-weight ratio and excellent corrosion resistance, titanium and its alloys are widely used in medical instruments and biomedical implants. According to the development history and long-term clinical feedback of medical titanium alloys, its future focus continues to be the in vitro biocompatibilities and mechanical compatibilities. The ultimate aim is to improve its in vivo biological safety and persistent service. Pure Ti and Ti3Al2.5V were chosen to make dental implants, which bear less stress in mouth, in the 1950s [1]. Since the end of the 1970s, a higher strength titanium alloy Ti6Al4V has been extensively applied in the medical field, such as hip joint, cardiac valves and artificial bones [2]. Ti6Al7Nb and Ti5Al2.5Fe were developed in the corresponding period but it was found that the elements aluminum and vanadium are harmful to the human body. In addition, the large difference between these titanium alloys (more than 100 GPa) and bones (3–40 GPa) causes stress shielding [3]. Because of the stress shielding, clinical images show that the prosthetic loosening, osteonecrosis and bone degeneration usually happen. In recent decades, a kind of titanium alloy with no toxic element, high strength and low elastic modulus has been the main subject for researchers. Till now, nearly 20 new titanium alloys have been successfully developed. They are called new β-type titanium alloys. The reason is that the elements molybdenum, thallium, niobium and so on, are stable elements of β-phase. The tensile properties are listed in Table 1. Ti3Zr2Sn3Mo25Nb (TLM) is one of these materials, and it has a low elastic modulus of approximately 45 GPa. So it is anticipated that TLM will play a role in the future.

Titanium and its alloys have been qualified in many fields, including space engineering, ocean engineering and biomedical engineering. However, high friction coefficient, poor wear resistance and low hardness limit its application [11]. For many aerospace components, shot peening is a standard finishing process, because of the compressive residual stress and strain-hardening induced by this treatment [12]. The rising of surface distortion and roughness is detrimental to the durability. To improve the fatigue strength of titanium alloys, other surface modification methods are tried by means of self nanocrystallization.

Surface self nanocrystallization (SSN) by mechanical process transforms the surface coarse grains of a bulk material into nano-sized grains by severe plastic deformation (SPD). These mechanical processes include surface mechanical attrition treatment (SMAT) [13], ultrasonic shot peening [14], laser shock peening [15], ultrasonic surface rolling processing [16], ultrasonic nanocrystal surface modification (UNSM) [17] and ultrasonic cold forging technology [18] In summary, the high energy of ultrasonic, laser and squeezing causes severe plastic deformation. It is reported that the mechanical properties can be improved by these methods [12,13,14,15,16,17,18].

In general, fatigue cracks usually initiate from the surface and subsurface. Most failures are sensitive to the micro-structures and the topography of the surface. So that, optimizing the surface of metals might enhance their overall performance. The grain refinement mechanisms of metals have been widely studied by researchers, including copper, steel, aluminum alloy, titanium alloy and so on. The reason nanostructured surface layers are generated from coarse-grains involves the dislocations, twinning and the development of grain boundaries with high angle misorientation [19]. It is accepted that the lattice structure and the stacking fault energy (SFE) decides the plastic deformation behavior and the dislocation in metals and alloys.

In the present work, a nanostructured surface layer was achieved by means of ultrasonic impact on TLM titanium alloy, which was processed with three different numbers of impacts per mm^2^. The micro-hardness, residual stress, grain size and crystal orientation were measured. The effect of ultrasonic impact on the rotating-bending fatigue behavior of TLM was also investigated. The stress intensity factor based on fish-eye model is calculated to analyze the mechanism of its fatigue characteristic.

## 2. Experimental Procedures

The test specimen in this investigation was a β-phase titanium alloy TLM titanium alloy shaft with the following chemical composition (mass %): Mo-3.1, Zr-3.07, Sn-2.09, Nb-24.8, C-0.015, O-0.16, N- <0.008, H-0.003 and balance Ti. All the specimens were treated with solid solution-aging (750 °C/30 m + 510 °C/4 h + furnace cooling). The mechanical properties of TLM are listed in Table 2. The tensile test was operated by a universal mechanical tester (CRIMS-DNS400, China Sinotest Sci. & Tech. Co. Ltd., Nantong, China).

The principle of the ultrasonic impact process uses ultrasonic vibratory energy at a frequency of 30 kHz, and tens of thousands of strikes per mm^2^ are applied to the material surface as constant pressure [20]. These strikes generate severe plastic deformation on the surface and induce a crystal refined surface layer till nano-sized. Compared with UNSM which is introduced by Pyun [21], the main difference is that the surface is processed without static load here. The ball tip has a diameter of 8.00 mm which is bigger than the one used in UNSM. It is made of cobalt with a tungsten carbide coating. The vibration amplitude was 30 μm. The numbers of vibration strike are 24,000 times/mm^2^, 36,000 times/mm^2^ and 48,000 times/mm^2^, respectively. Four groups of specimens, referred to as untreated, UI-24000, UI-36000 and UI-48000, were prepared.

Cross-sectional observations of the core area was performed with a scanning electron microscope (S-3400N, Hitachi, Nantong, China). After polishing, the specimens were finally etched in Kroll’s reagent (HNO_3_:HF:H_2_O = 3:6:90, vol%). Transmission electron microscopy (TEM) investigations were carried out with a FEI Tecnai G2 F20 S-TWIN. The specimen for TEM was prepared by focused ion beam (FEI Helios Nanolab 600i), to a depth of 0.5 μm. The orientation, length and density were characterized by electron backscatter diffraction (EBSD, Leica EM RES 102). The surface of the specimen was fine polished with argon ion.

The microhardness was measured with a Vickers hardness tester (CMM-20E, Changfang, Chengdu, China) with a load of 50 g and duration of 20 s. The hardness test was performed along three lines. X-ray diffraction (XRD) was used to measure the residual stress using Rigaku X’pert pro MPD. The Cu Kα radiation (λ = 1.54184 Å) was used, and the diffraction lattice plane (213) was examined within a range 2θ of 113° to 116°.

To investigate the effect of the nanostructured surface layer on the fatigue behavior of TLM, rotating-bending fatigue test on the four groups of samples (untreated, UI-24000, UI-36000 and UI-48000) were conducted at ambient temperature using an Ono fatigue test machine (Shimadzu, Fukuoka, Japan). The load is applied as four point bending. The frequency was 50 Hz. The dimensions of the fatigue sample are illustrated in Figure 1. The fracture surfaces were examined using a scanning electron microscope (GeminiSEM 300, Zeiss). Energy-dispersive X-ray spectroscopy (EDX) was also used to detect the composition of any inclusions which induced the inner crack initiation with it.

## 3. Results and Discussion

### 3.1. Observation of the Severe Plastic Deformation Layer

The microstructure characterizations of the SPD layers are usually described by means of TEM. Furthermore, it is generally accepted that nanocrystals can be obtained with the ultrasonic surface strikes [22]. The microstructural changes of the TLM titanium alloy surface are shown in Figure 2. It indicates that high density dislocations and dislocation cells are generated by ultrasonic strikes. Dislocation cells are large and have thin walls composed of tangled dislocations. Continuous slip band are observed on the β-phase crystal obviously. In Figure 2a, it shows that an amorphous region is formed underneath the surface and the grains are refined to nanometer scales. The average size is about 30 nm. According to Figure 3, the original grain size of the base material is approximately 50 μm, the microstructures are observed as isometric crystals. It is demonstrated that the parameter combinations without static load are enough to generate nanograins in the surface of the TLM titanium alloy. It is introduced that the dynamic load of ultrasonic strikes is about two times that of static load [23], severe plastic deformation layers are usually shown with rheological trend. However, the legible strain flow was not observed because static load was not introduced here. It is reported that twinning exists in α-titanium because of the low hexagonal symmetry [24]. Hexagonal close packed (hcp) α-phase has an SFE higher than 300 mJ/m^2^, body centered cubic (bcc) β-phase theoretically has 12 slip directions. Thus, for this β-type TLM titanium alloy, the mechanism of grain refining in TLM titanium alloy is mainly dislocation motion. Because of the large amount of nanocrystals, the SAED pattern of the SPD layer is nearly concentric annulus instead of an hcp or bcc lattice.

### 3.2. EBSD Analysis

After a scanning from surface to the depth of 30 μm with a small step width of 0.035 μm, it was clear that crystals begin to be continuously distinguished by EBSD up to 20 μm. This means that the depth of nanocrystal layers are not more than 20 μm. The misorientation of the boundary is characterized by values of 2° to 60°. Figure 4a shows the area examined by EBSD, which is a square with a depth to center of 100 μm. Figure 4b shows the unique grain color map with the inverse pole figure. Each grain is assigned a color to distinguish it from neighboring grains. Because this square starts from the depth of 25 μm only a small quantity of grains with the size of less than 10 μm are observed. The average size of the grains on the right side, are larger than those on the left. The pink spots indicate that dislocation glide happens in that position. Figure 4c shows the inverse pole figure analysis and the change in color in each grain corresponds to that in crystal orientation. For the plasticity induced by tensile testing in the beta-titanium alloy, twinning is observed as the main reason [25]. The primary causes of severe plastic deformation of TLM are the dislocation motion and a little of the lattice rotation. From the orientation information, it can be seen that there is obvious preferred orientation in the grains. The c axis of grain is mainly in Y direction.

### 3.3. Micro Vickers Hardness Distrubution

The average hardness along the depth from the surface to 400 μm is depicted in Figure 5. Where, the hardness was measured perpendicular to the cross-sectional while it was tested vertical to the surface. It is about 238 HV of the untreated TLM base material. The micro-hardness of UI-24000, UI-36000 and UI-48000 is 288 HV (about 21% improvement), 294 HV and 291 HV, respectively. The hardness after surface impact rapidly decreases to 200 μm and then decreases gradually to the core. It is well known that the hardness and the yield stress relate to the grain size according to the Hall–Petch theory. It is reported that the subgrain size will not change while the dislocation multiplication rate is balanced [19,26]. At this time, the maximum surface hardness will not increase any more. In this paper, each point is subjected to the ultrasonic impacts for two seconds without static load. The surface hardness would be anticipated to be more if the strike number per unit was bigger and the static pressure was adopted.

### 3.4. Residual Stress

It is well known that high compressive residual stress can be induced by means of ultrasonic impact [27,28]. The surface residual stress of untreated specimens is a tensile stress of 12.05 MPa. For the TLM specimens subjected to ultrasonic surface impact, the residual stresses below the depth of 20 μm were measured. Table 3 lists the residual stress measurement results. It is observed that compressive residual stress is induced at the surface of the TLM specimens. As the strike number per unit increases, the value increases synchronously. The compressive residual stress of UI-24000, UI-36000 and UI-48000 are 247.84 MPa, 273.60 MPa and 288.52 MPa, respectively. The stress values here are less than that of materials treated by UNSM with static load, such as SCM 435 [29]. Compressive residual stress is an important factor for increasing the fatigue resistance. It effects the position and the shape of inner crack initiation. It is reported that the high compressive residual stress of TC4 which is treated by UNSM rapidly decreases from the depth of 50 μm to 200 μm [5]. Thus, cracks are initiated easily in this zone without the nanograins and the compressive residual stress.

### 3.5. Tensile Properties

Figure 6 shows a tensile stress-strain curve for TLM titanium alloy before and after ultrasonic surface impact treatment. Because the differentiation of four tensile curves is tiny, only the curves of untreated and UI-48000 are given. The sample of UI-48000 has a nanostructured surface layer. As mentioned above, the thickness of this layer is less than 20 μm. Both the samples have the same geometry. The yield strength increases by 3.17%, while the ultimate tensile stress is increased by about 2.36% (from 721 MPa to 738 MPa). It is reported that the tensile stress of 316 L stainless steel is increased by about 13%, while the yield strength increases from 280 MPa to 550 MPa after SMAT. The reason is the phase transformation of martensite [23].

Elastic modulus *E* is a parameter which concerns the cohesion of atoms. It is defined as the slope of its stress-strain curve in the elastic deformation region. Elastic modulus of nanocrystalline Fe, Cu, Ni and Cu-Ni alloys are lower because of a relatively large volume of pores [30]. From the partial enlarged detail, it can be seen that the elastic modulus of TLM titanium alloy has a minor decrease. Even though the base material gives the most contribution of tensile property, the changes of the mechanical parameters are certainly linked to the grain refinement.

### 3.6. Fatigue Characteristics

The fatigue characteristics of the TLM titanium alloy subjected to ultrasonic surface impact are shown in Figure 7. Specimens that did not encounter failure are shown as run-outs. Inner cracks are marked with vertical bars. After the ultrasonic surface impact process, most cracks transform into inner cracks while the fatigue lives are longer than 10^6^ cycles. Contrastingly, cracks initiate at the surface to all the un-treated specimens. From the S-N curves, it is evident that ultrasonic strikes improve the 10^7^ cycles fatigue strength of TLM. The fatigue strength of UI-24000, UI-36000 and UI-48000 are increased by 13.1%, 15.8% and 23.7%, respectively. Table 4 lists the enhancement of the strength and fatigue limit by ultrasonic surface impact. After surface strengthening by ultrasonic impact, the fatigue performance of bending fatigue specimens was improved and the 10^7^ cycles fatigue strength of treated specimens was within 0.50 to 0.65 σ_b_. The fatigue data of TLM confirms this data. For some austenite stainless steels with the phase transformation of martensite, the fatigue strength will be further improved [31].

Figure 8 and Figure 9 are the SEM micrographs of the fatigue fracture surface of TLM titanium alloy with fatigue lives of more than 10^6^ cycles. Some internal cracks are investigated in this test. All the cracks initiate from the surface of S45C specimens after UNSM which are subjected to a rotating-bending fatigue test [32]. Maximum normal stress in bending is on the surface. Comparing the surface topography of UNSM and which is applied here (as reported by Ao [20]), finer strike tip will cause microcracks easier. The surface integrity influences the fracture crack behavior. To surface self nanocrystallized (SSN) specimen, a nano-structured layer is achieved on the surface. Deeper down there is a refined structured layer consisting of submicrometer-sized crystallites separated by either grain boundaries or sub-boundaries [19]. Then a deformed coarse grain layer forms at a certain depth. In the zone of both the refined structured layer and the deformed coarse grain layer, the lattice distortion and the crystal slip have not balanced to the saturated state. Noticeably, the inner crack cores are at this zone [35]. In addition, the compressed residual stress decreases quickly, the hardness is closed to the core and plasticity and tenacity are weakened in this area [5]. Therefore, indented slip traces of crystals are obvious in the crack initiation. The crystal slip is the main way of the growth of microcracks in the crack initiation zone. It is considered that the facet zone has many small facets due to the slips. Crack cores are usually formed at the junction of crystals instead of inclusions. Thus, it is deduced that the triple grain boundary junction shall be the initiation of the inner fatigue crack as illustrated in Figure 10. For the rotating-bending fatigue test, the maximum normal stress is at the surface. The cracks of all the untreated specimens are generated at the surface. Thus, the crack initiation mechanism is transformed with the application of ultrasonic surface impact. Nanograins prevent the initiation of surface crack. Microcracks which are generated by the plastic overflow after ultrasonic surface strike are observed to be surface defects [5]. Because of the surface maximum bending normal stress and the surface micro defects, some specimens have surface crack initiation with a fatigue life of more than 10^6^ cycles.

As it is known that fish-eye-cracks are divided into three areas: inclusion area, facet area and flat area. To most iron-based materials, the first area is the non-metallic compound inclusion in the center of a fish-eye. It is the triple grain boundary junction of the crystal slip to TLM instead. The second is the facet area with a rough surface, as marked in Figure 8a. The flat area is seen as a dark gray ellipse located around the facet area. The stress intensity factor, Δ*K*, relevant to the facet area can be calculated by Equations (1) and (2) [36],
(1)$$\Delta K=0.5\sigma_{at}{\left(\pi \sqrt{area}\right)}^{1/2}$$
(2)$$\sigma_{at}=(d-2h)\sigma_{a}/d$$
where *σ_a_* is the nominal stress amplitude at the surface, *σ_at_* is stress amplitude at the inner crack core, *d* is diameter of specimen, *h* is depth of the crack initiation and *area* is the area of each part. For the TLM titanium alloy, the stress intensity factor calculated from the facet area Δ*K_facet_* is about 7.0 MPa·m^1/2^. The stress intensity factor Δ*K_facet_* and Δ*K_inclusion_* have different propagation mechanisms. For titanium alloys, the non-metallic compound inclusion is not often observed. The facet area can be delimited with the micro facets due to the slip of α-phase. From the Murakami equation, it is clear that Δ*K_facet_* is close to the threshold stress intensity factor Δ*K_th_*. Normally, Δ*K_th_* is the value to divide the initiation and the steady expanding of cracks according to the fatigue crack growth rate curve. The time of crack initiation usually lasts more than 95% of the whole fatigue life. To materials with inner cracks, Murakami considers that there is a confirmable maximum value of Δ*K_facet_* by multiple experiments. With the size of inner cracks and the stress intensity factor, the fatigue limit can be estimated by the inversion of the above equations.

## 4. Conclusions

Nanocrystals with a size of 30 nm are generated with the ultrasonic surface impacts. The primary causes of severe plastic deformation of the TLM titanium alloy are the dislocation motion and a little of the lattice rotation. Ultrasonic surface impact improves the micro hardness, the surface compressive residual stress, tensile strength and the fatigue strength of TLM. Specimens which are treated with the higher vibration strike number will have a better enhancement of mechanical properties. With the vibration strike number of 48,000 times per unit, the rotating-bending fatigue strength of TLM at 10^7^ cycles is 470 MPa (23.7% improvement). Fatigue cracks mainly initiate from the surface of the specimen before the fatigue life of 10^6^ cycles, while inner cracks appear at the boundary junction of the crystals after the fatigue life of 10^6^ cycles. The stress intensity factor of TLM is approximately 7.0 MPa·m^1/2^.

## Figures and Tables

**Figure 1 materials-13-02107-f001:**
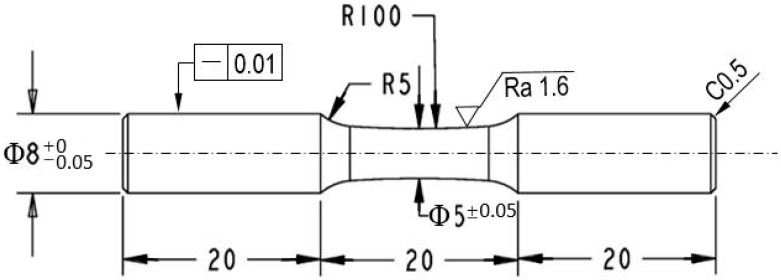
Geometrical dimensions and surface characteristics of the fatigue specimen.

**Figure 2 materials-13-02107-f002:**
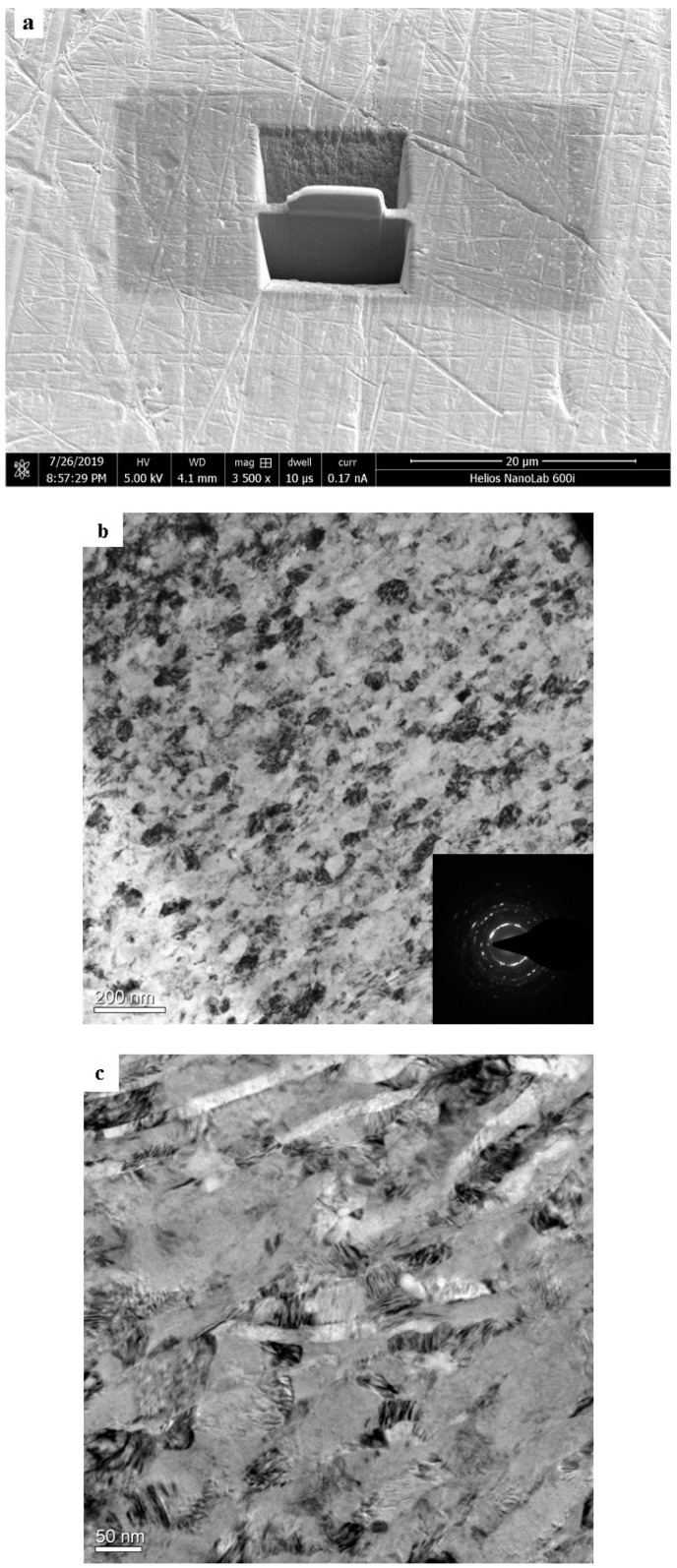
TEM observation of TLM with ultrasonic surface impact: (**a**) TEM sample prepared by FIB; (**b**) image and diffraction pattern; (**c**) dislocation walls.

**Figure 3 materials-13-02107-f003:**
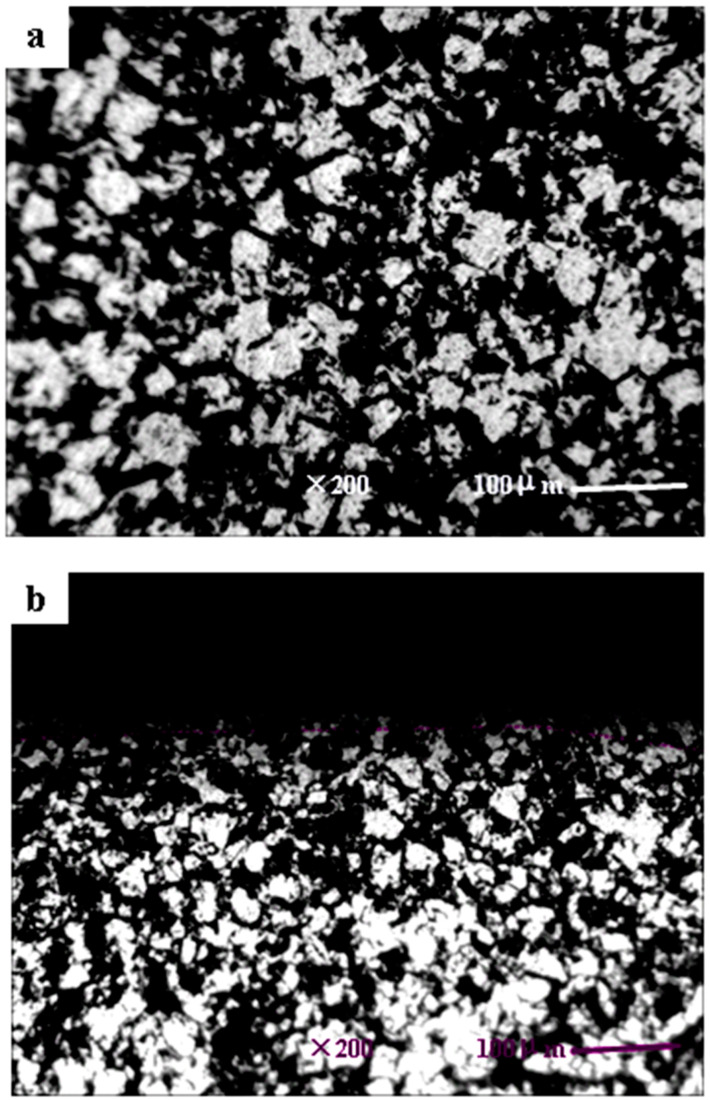
Optical micrograph of TLM titanium alloy: (**a**) Base material; (**b**) UI-48000.

**Figure 4 materials-13-02107-f004:**
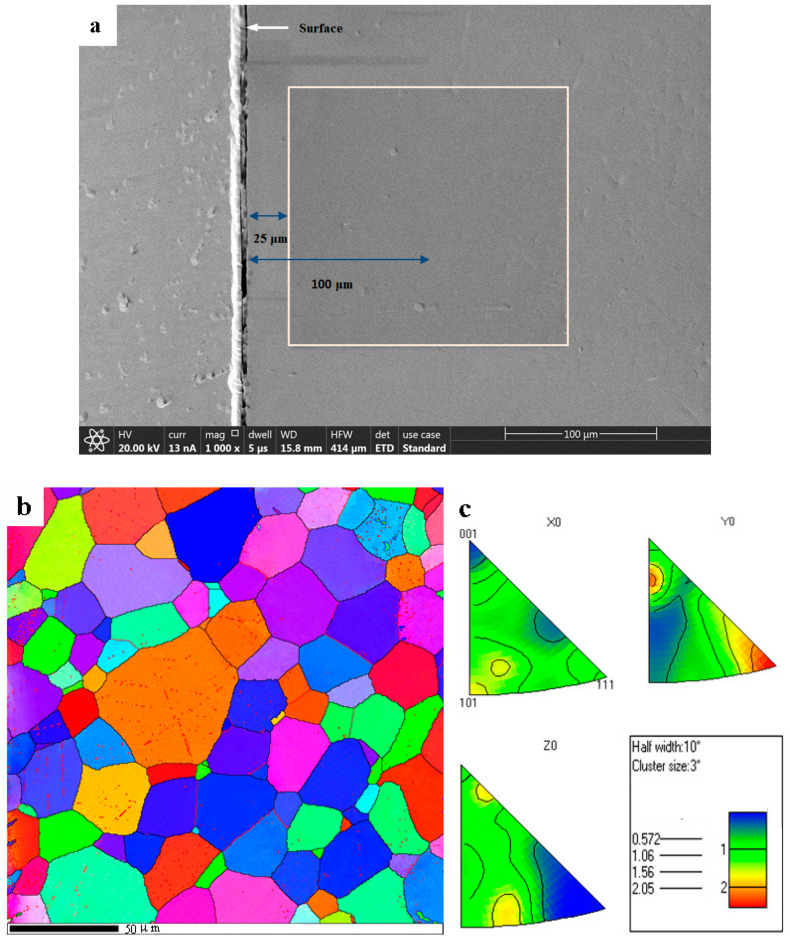
EBSD analysis results: (**a**) Scanning zone; (**b**) grain boundary map; (**c**) inverse pole figure.

**Figure 5 materials-13-02107-f005:**
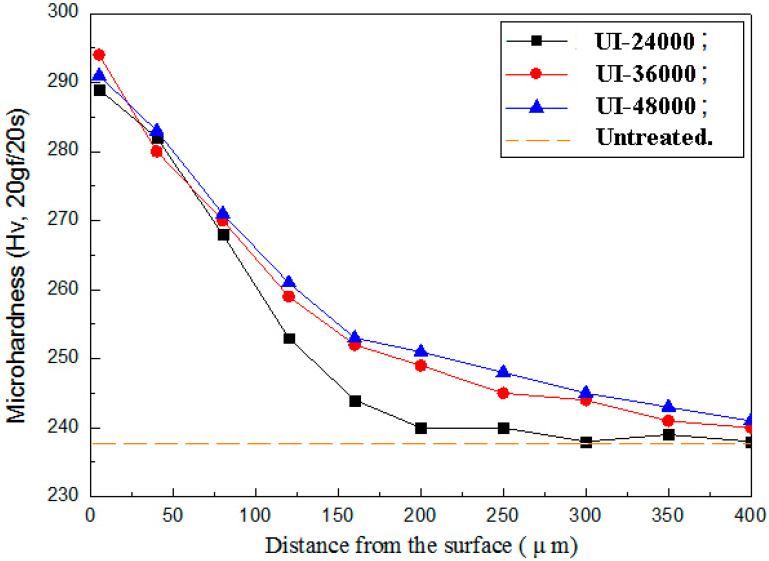
Distribution of surface hardness with ultrasonic surface impact.

**Figure 6 materials-13-02107-f006:**
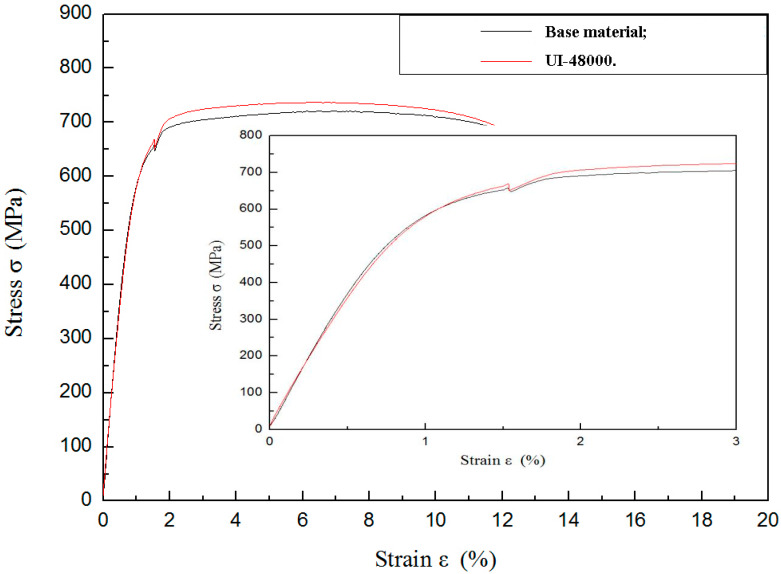
Tensile stress-strain test for TLM titanium alloy.

**Figure 7 materials-13-02107-f007:**
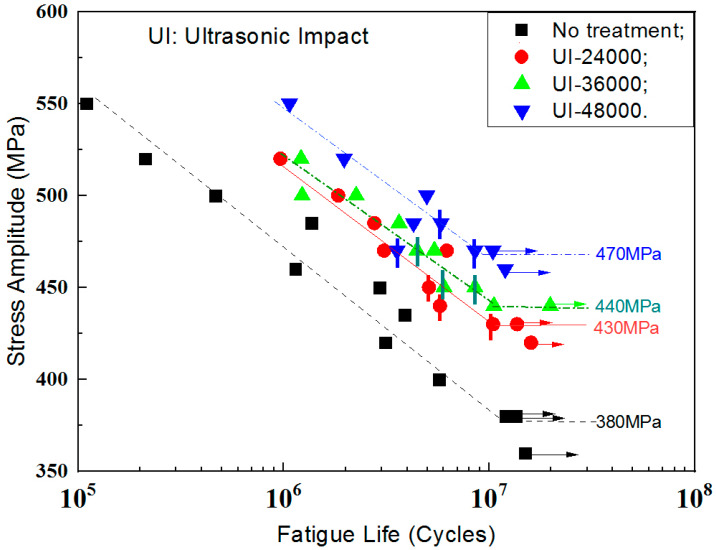
Fatigue S-N curves of TLM titanium alloy.

**Figure 8 materials-13-02107-f008:**
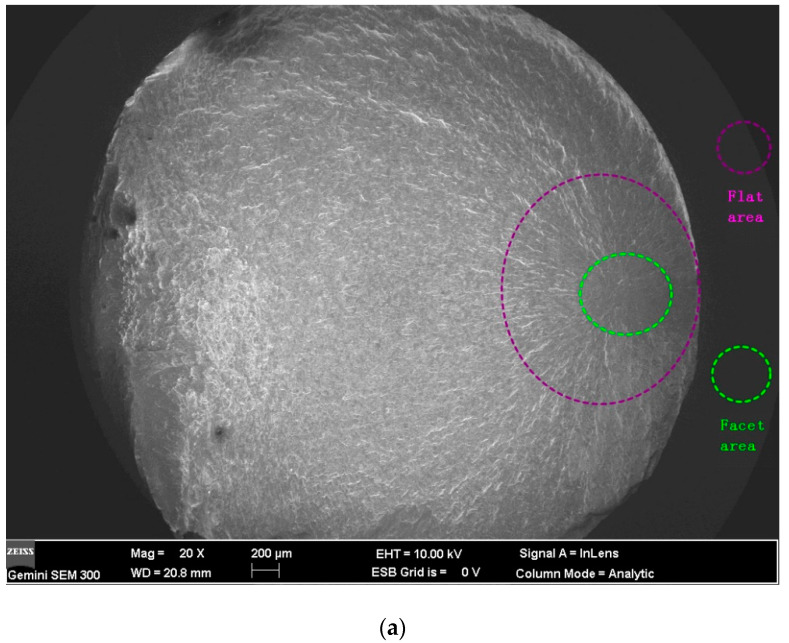
Fracture surface of TLM (UI-24000, 500 MPa, 1.86 × 10^6^ cycles): (**a**) Overall view of fracture surface; (**b**) slip trace of crystals in the crack initiation; (**c**) EDX analysis of crack core.

**Figure 9 materials-13-02107-f009:**
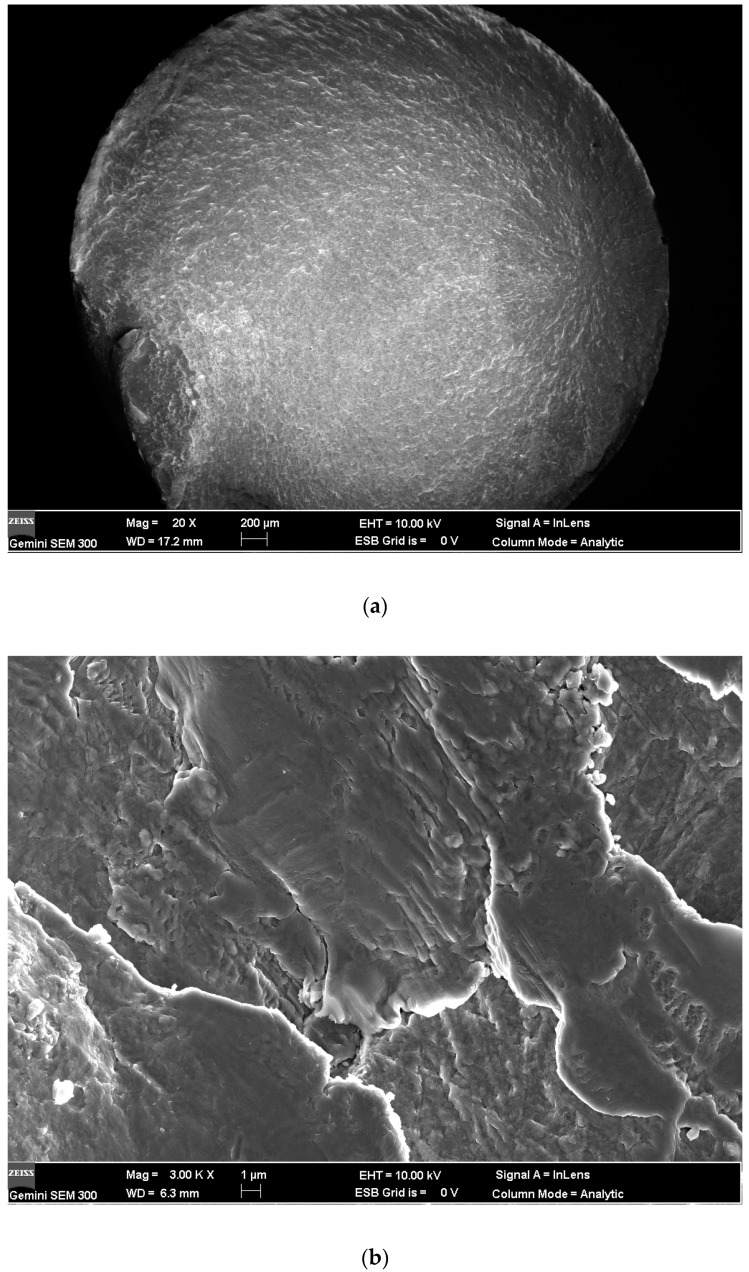
Fracture surface of TLM (UI-48000, 500 MPa, 4.97 × 10^6^ cycles): (**a**) Overall view of fracture surface; (**b**) slip trace of crystals in the crack initiation.

**Figure 10 materials-13-02107-f010:**
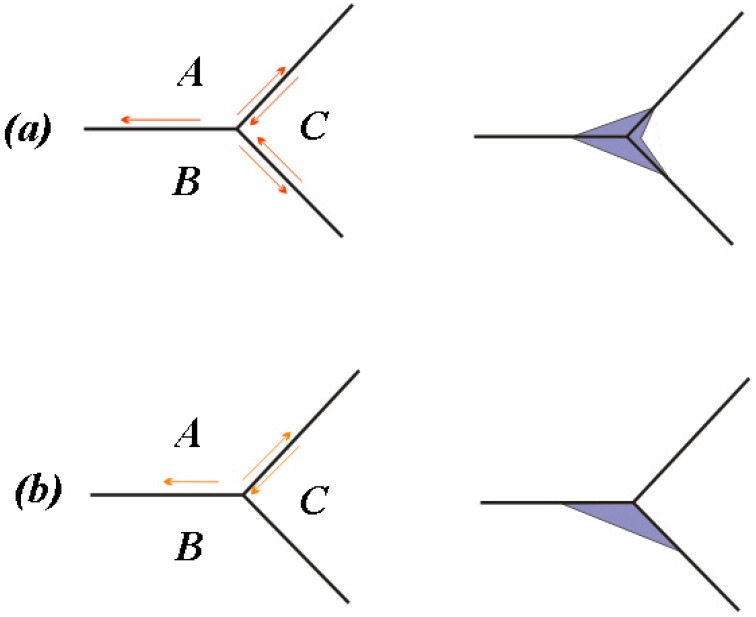
Illustration of the crack at the triple grain boundary junction of the crystal slip: (**a**) Slips on three grain boundaries; (**b**) Slips on two grain boundaries.

**Table 1 materials-13-02107-t001:** Mechanical properties of some titanium alloys for biomedical use.

Material	σ_0.2_ (MPa)	σ_b_ (MPa)	*δ* (%)	*Ψ* (%)	Elastic Modulus (GPa)
Pure Ti [4]	170–485	240–550	15–24	25–30	≈103
Ti6Al4V(annealed) [5]	820–870	900–930	6–10	20–25	110–114
Ti6Al7Nb [6]	880–950	900–1050	8–15	25–45	114
Ti13Nb13Zr(aged) [7]	830–910	970–1040	10–16	27–53	79–84
Ti15Mo(annealed) [8]	511	874	21	82	78
Ti12Mo6Zr2Fe(annealed) [9]	1000–1060	1060–1100	18–22	64–73	74–85
TLM(aged)	610–950	685–1050	17–23	70–71	45–81
Ti24Nb4Zr7.9Sn(aged) [10]	800–1100	850–1150	15	——	42–82

**Table 2 materials-13-02107-t002:** The mechanical properties of TLM titanium alloy.

Heat Treatment	σ_0.2_/MPa	σ_b_/MPa	Elongation/%	Reduction of Area/%
Solution and aging	567	721	19.5	71

**Table 3 materials-13-02107-t003:** Residual stress of TLM titanium alloy.

Processing Method	Untreated	UI-24000	UI-36000	UI-48000
Residual stress (MPa)	+12.05	−247.84	−273.60	−288.52

**Table 4 materials-13-02107-t004:** Enhancement of rotating-bending fatigue limit by ultrasonic surface impact.

Material	Heat Treatment	Yield Strength σ_0.2_ (MPa)	Tensile Strength σ_b_ (MPa)	Fatigue Limit at 10^7^ Cycles (MPa)	
				Before	After
S45C [32]	Annealed	490	690	300	400
SUS304 [31]	Hot rolling	205	520	280	520
SCM435 [29]	770 °C × 3 h + 680 °C × 10 h	836	991	500	650
ATI 718 plus alloy [27]	788 °C × 8 h + 704 °C × 8 h	1200	-	740	850
Inconel 718 alloy [33]	Annealed 954 °C × 30 m	829	1117	-	≈700
A6061 [34]	T6	276	310	130	180

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
