# Peer review of "Effect of Ultrasonic Surface Impact on the Fatigue Properties of Ti3Zr2Sn3Mo25Nb"

_materials, 2020, doi:10.3390/ma13092107_

Round 1
Reviewer 1 Report
The authors studied the effect of ultrasonic-impact-induced surface modification on the fatigue behavior of a Ti alloy. Three different conditions are compared, giving each a different amount of residual compressive stress at a distance of 20 um from the surface. An improvement is observed for each condition with respect to the untreated material, and is more for the treatment leading to larger residual compressive stress measured at 20 um from surface. A different location for fatigue crack nucleation is observed in some of the treated materials, for the longer fatigue lives.
Overall, this is a nice, comprehensive set of experimental data, including TEM, SEM observations, XRD residual stress measurements, and hardness measurements, in addition of S-N curves. However, the English of the manuscript needs to be improved, as there are sections that are difficult to understand. The analysis needs some clarification as well.
- Lines 122-125. Unclear what the authors mean. Please rephrase.
- Fig 4 and 5. If the grains are similar to the untreated specimens below 25 um from the surface, why are the hardness values larger for the treated specimens even 100 um below the surface?
- Table 3 and Fig 5. Is there a relationship between the residual stress at -20um, and the hardness profile? What does a larger compressive stress mean in term of the extent of surface modification? Smaller grains? Larger layer with smaller grains?
- Table 3. Can the authors comment on the residual stress profile from the surface to 100-200 um deep? How does that change for the 3 conditions?
- Lines 210-226: very difficult to understand. Also, the authors appear to conclude that the subsurface cracks initiate at triple points. What proof do they have exactly? It appears to me to be a hypothesis, and not a conclusion.
- Fig 7: many of the treated specimens with Nf > 10^6 cycles have fatigue cracks initiating at the surface (no vertical bar on graph). How do the authors explain that?
- Ines 241-250: can the authors give the details of their calculation to get 7 MPa m1/2? How many specimens? What values for d, h, area, etc? Can they explain how this value can be used to predict fatigue limit? Can they explain the difference between DKfacet and DKinclusion?
Author Response
1.Beta titanium alloy is not all of beta phase (light part in metallographical photo), there is also some alfa phase. So both of them shall be simply mentioned. 2.The hardness diagram shows the average value of each depth with three points tested. The enhancement of UI-48000 is about 8%. Comparing with 304 stainless steel (surface hardness is three times of base material), S45C(two times of base material). It is even negligible. The hardness is related to the grain size, the grain size does not have a big difference as shown in EBSD photo. 3.There is not a certain relationship (such as equation or theory) between hardness and residual stress. The residual stress values are tested from plane specimen with ultrasonic strike. X-ray can just arrive the depth of nearly 15~20 μm in metals. The voids between nano crystals shall be compressed like springs and air cusion. In my opinion, this is the reason of the residual stress. 4.The values of residual stress are similar. The reasons are: without static load (static load plus ultrasonic strike can generate a compressive residual stress of 1500MPa), it is not tested immediately after the treatment(residual stress releases). 5.EDX was also investigated, no inclusion is found. Slip bands around the crack inclusion along two or more directions are observed, and a hole is formed in each inner-crack. This means at least three crystals are contributory. So that, the conclusion is made. 6.The nanocrystal surface modification by means of mechanical method such as peening, rolling, striking shall bring surface damage more or less. Other specimens are with surface cracks due to the surface micro-cracks. 7.From the theory which given by Prof. Murakami(Kyushu University), DKfacet value of each metal has a unique section. This is the character of materials. Oppositely, DKinclusion is discrete .Reviewer 2 Report
Please operate the minor corrections:

Author Response
The mentioned (nit limited to) marking errors are corrected. The contrast ratio of etallographical photos are changed. Thank you very much.Reviewer 3 Report
The paper deals with the fatigue behaviour assessment of TLM Titanium alloy subjected to ultrasonic surface impact (the treatment involved three different values of vibration strike number). The discussion on fatigue behaviour was supported by residual stress, micro-hardness and quantitative microscopy analysis.
Even if the research is interesting and provides new tools to improve the fatigue resistance of Titanium alloys, there are some open points and some confusing parts that must to be explained and improved in order to increase the quality of the manuscript.
- In my opinion, the choice of the values of test parameter three vibration strike numbers has to be motivated.
- Since the paper is focused on fatigue behaviour assessment it is essentially to provide a wider discussion and critical analysis of the fatigue behaviour of the investigated alloy.
- The section 2 , requires more specifications:
- Include the micrographs of base material to better understand the microstructure
- Micro-hardness tests: did the tests were performed along a line?in a matrix? Please, specify if the tests involved a single point or a pattern. Include the number of tests performed per sample.
- Bending Fatigue tests: which kind the fatigue tests were performed? Constant amplitude? load or displacement control? Standard? Number of samples tested? How many replication for each one test?
- The authors presented some results on the tensile tests. Why tensile tests aren’t described in the section 2?
- Why just a number of 2 samples for tensile tests? Please add more information. It is not clear if the curves of fig 6 are averaged or refer to just one sample per typology. Please specify
- It is not clear why the author refers to some stainless steel in the text. Please refer to literature referring the alloys you tested, otherwise specify the link.
- 7: how many test replication for each point of the SN? Please include the fitting of the data, it can support in understanding the differences ultrasonic surface impact treatments. How can the authors justify the little difference in SN curve between 24000 and 36000? Please add a table containing the estimation of endurance limit resulting from fatigue tests (Fig. 7)
Author Response
1.yes, I agree with you. 2.Without static load on the surface, SPD layer is not legible. It is reported that the depth of SPD layer enhanced as increasing the static load (SUH C, YOON S, JANG J , et al. Very high cycle fatigue characteristics of SCM435 under load variation by ultrasonic nanocrystal surface modification treatment[J]. The Korean Society of Mechanical Engineers, 2009, 11: 66-71.). 3. The hardness diagram shows the average value of each depth with three points tested. The enhancement of UI-48000 is about 8%. Comparing with 304 stainless steel (surface hardness is three times of base material), S45C(two times of base material). It is even negligible. The hardness is related to the grain size, the grain size does not have a big difference as shown in EBSD photo. 4.The corresponding author has been a visiting scholar in Kyushu university for half a year in 2019. Nearly 160 fatigue test specimens are prepared (10 groups). But one piece one time can be operated on the Ono fatigue tester(four point bending), the fatigue could not be put across. The fatigue limit at 10E7 cycles was given priority to be the aim. 5.It is added. 6.Only two specimens (untreated VS UI-4800) has a better result. The nanocrystal surface modification by means of mechanical method such as peening, rolling, striking shall bring surface damage more or less. Other specimens are with surface cracks due to the surface micro-cracks. 7. (YASUOKA M, WANG P, ZHANG K, et al. Improvement of the fatigue strength of SUS304 austenite stainless steel using ultrasonic nanocrystal surface modification[J]. Surface & Coatings Technology, 2013, 218: 93-98.) It has been a reference. 8. Weak phase change is also observed in TLM alloy. So it is mention sometimes in the manuscripts. Thank you very much.Round 2
Reviewer 1 Report
In my opinion, the changes the reviewers made to the manuscript are superficial, and the response to the comments are insufficient. I also have a hard time understanding the responses due to the poor quality of the English, which certainly doesn't help.
My recommendation is therefore to reject this paper.
Author Response
Thank you very much for the friendly comments. Hope that everything is all right during the outbreak of COVID-2019.
1.The correction was given in the mentioned lines.
2.The hardness diagram shows the average value of each depth with three points tested. The enhancement of UI-48000 is about 8%. Comparing with 304 stainless steel (surface hardness is three times of base material), S45C(two times of base material). It is even negligible. As introduced in the manuscript, the second test depth of hardness is 30 μm where the grain size is similar to the base material. The surface hardness is tested vertical to the surface instead of the cross section. It is possible that the indentation has reached the polycrystalline area. Thus the effect of the enhancement of hardness is imperfect.
3.There is not a certain relationship (such as equation or theory) between hardness and residual stress. Plastic deformation under the ultrasonic strike causes the residual stress. In fact, surface compressive residual stress shall transform to be tensile stress because of the self-balance of inner stress. A large compressive stress means the severe plastic deformation. The grain can’t be refined infinitely. We can just describe this plastic deformation is saturated or not. The stress field under the ultrasonic strike is limited.
- The analysis of the residual stress along the depth was added in this part. As you know, the test of residual stress distribution layer by layer is difficult. The tiny distinction between three conditions is because of the different rate of plastic deformation and the number of nanograins. All the three groups are far away from the balanced saturated conditions.
5.EDX analysis of crack core was added, no inclusion is found. The mentioned part has been rewritten in the abstract, conclusion and the corresponding part of fracture surface analysis.
6.The nanocrystal surface modification by means of mechanical method such as peening, rolling, striking shall bring surface damage more or less. Because of the surface maximum bending normal stress and the surface micro defects, some specimens have surface crack initiation with a fatigue life more than 106 cycles. It is added.
- More than 20 specimens have inner cracks. The stress intensity factors are calculated. The given 7 MPa m1/2 is the maximum approximate value due to Murakami’s theory. It is considered to be similar with DKth. this is added in the manuscript. And the explain of how to predict fatigue limit is simply introduced.
Best wishes.
Author Response
Thank you very much for the friendly comments. Hope that everything is all right during the outbreak of COVID-2019.
1.The introduction of tests are introduced. The EDX result is added.
2.Without static load on the surface, SPD layer is not legible even on the cross section of UI-48000. The resolution ratio of optical pictures is increased to be better. It is reported that the depth of SPD layer enhanced as increasing the static load (SUH C, YOON S, JANG J , et al. Very high cycle fatigue characteristics of SCM435 under load variation by ultrasonic nanocrystal surface modification treatment[J]. The Korean Society of Mechanical Engineers, 2009, 11: 66-71.).
3.The hardness diagram shows the average value of each depth with three points tested. They are performed along three lines. The enhancement of UI-48000 is about 8%. Comparing with 304 stainless steel (surface hardness is three times of base material), S45C(two times of base material). It is even negligible.
4.The corresponding author has been a visiting scholar in Kyushu university for half a year in 2019. Nearly 160 fatigue test specimens are prepared (10 groups). But one piece one time can be operated on the Ono fatigue tester(four point bending), the fatigue could not be put across. The fatigue limit at 10E7 cycles was given priority to be the aim. The bending fatigue test is operated with a calculated load to control the maximum bending stress.
5.It is added in section 2.
6.Because the differentiation of four tensile curves is tiny, only the curves of untreated and UI-48000 are given. The effect of selfsurface nano grains by SMAT was described by Lu(K. Lu, J. Lu. Nanostructured surface layer on metallic materials induced by surface mechanical attrition treatment. Mater. Sci. Eng. A. 2004, 375-377: 38-45.). I think this might be universal phenomenon. But it can’t be identified in this paper.
7.(YASUOKA M, WANG P, ZHANG K, et al. Improvement of the fatigue strength of SUS304 austenite stainless steel using ultrasonic nanocrystal surface modification[J]. Surface & Coatings Technology, 2013, 218: 93-98.) It has been a reference.
8.Weak phase change is also observed in TLM alloy. So it is mention sometimes in the manuscripts.
Best wishes,
